# Study on the Corrosion Behavior and Mechanism of ER8 Wheel Steel in Neutral NaCl Solution

**Cheng-Gang He [1], Zhi-Bo Song [1], Yao-Zhe Gan [1], Rong-Wei Ye [2], Run-Zhi Zhu [1,3], Ji-Hua Liu [1,*] and Zhi-Biao Xu [1]**

[1] School of Railway Tracks and Transportation, Wuyi University, Jiangmen 529020, China; hechengan0522@wyu.edu.cn (C.-G.H.); songzb1998@126.com (Z.-B.S.); woshiganyaozhe@163.com (Y.-Z.G.); hangcheng_z@163.com (R.-Z.Z.); 13547922554@163.com (Z.-B.X.)

[2] Comprehensive Training Center of Modern Industrial Production Technology, Wuyi University, Jiangmen 529020, China; jmyrw@126.com

[3] Foshan Institute of Intelligent Equipment Technology, Foshan 528234, China

[*] Correspondence: ljh214913@163.com; Tel.: +86-750-3299-196

**Abstract:** This paper analyzed the corrosion behavior and corrosion performance of ER8 wheel steel through a full immersion test. The average corrosion rate of the ER8 wheel specimen in 2.0% NaCl solution shows a gradual increase over the whole corrosion cycle. Although the corrosion rate showed fluctuations at 3.5% and 5.0% concentration before 576 h, the corrosion rate also showed a steady increase after 576 h. The corrosion rates of specimens at different concentrations after 2160 h were over 0.12 mm/year. With increasing immersion times or concentrations of NaCl solution, the coverage area of the corrosion products dominated by iron oxides gradually increased, and the corrosion products on the surface became denser. The corrosion products were primarily $\gamma$-FeOOH, $\alpha$-FeOOH and $Fe_3O_4$. As the density of the surface corrosion products increased, cracks and holes appeared on the surface of the rust layers, which made the rust layer unable to protect the substrate from further corrosion. After removing the corrosion products, pitting corrosion appeared on the surface of the substrate. The radius of the capacitive reactance arc gradually decreased with the increasing immersion time. The impedance modulus in the low-frequency region decreases and then increases with increasing NaCl solution concentration, which is the highest in 3.5% NaCl solution. $I_{corr}$ increased with an increasing $Cl^-$ concentration, which was similar to the mechanism of catalytic electrolysis due to $Cl^-$. The specimens with rust layers have worse corrosion resistance when the immersion time is extended. The corrosion product did not protect the substrate but accelerated the corrosion process.

**Keywords:** ER8 wheel steel; neutral chloride solution; corrosion rate; electrochemical testing; corrosion mechanism

## 1. Introduction

Carbon steel is an indispensable material for the development and survival of society and is widely used in industrial production, construction, and national defense [1–4]. Wheel material is a typical medium- and high-carbon steel material. After year-round exposure to the outdoor environment, changes in environmental conditions causes severe stresses in the wheel, and temperature and humidity promote the corrosion of wheel steel. The corrosion of iron and steel appears in all walks of life, causing great harm to our normal life, the environment, and the basic resources of countries. The annual steel loss caused by corrosion and rust accounts for approximately 10~20% of the annual steel produced, which brings about many economic losses. With the rapid development of high-speed railways, research on wheels has become important both domestically and overseas, and the corrosion resistance of train wheels is the same as its strength and toughness, which are related to the safe operation of trains [5,6].

There has been much research indicating that the main components of the rust layer of carbon steel exposed to the atmosphere for a long time are γ-FeOOH, α-FeOOH and $Fe_3O_4$ [7–9]. Some research has found that the outer rust layer normally contains an oxidizer, which is thought to be iron oxide [10]. In a littoral environment, corrosion products act as a strong oxidant that alternates with oxygen as the reducing agent in the wheel corrosion reaction [11].

To date, most research on wheel and rail materials has been related to friction and wear, whereas research on corrosion is often limited [12–14]. In addition, many studies have shown that $Cl^-$ has an impact on the corrosion behavior of carbon steel. The evolution of the rust layer has also been studied extensively, but there are few studies that also relate to train wheel steels [15,16]; therefore, it is of great significance to study the corrosion rate and corrosion behavior of ER8 wheel steel in environments with different chlorine concentrations to predict the service life of wheels.

The present study focuses on clarifying the corrosion behavior of ER8 wheel steel and the evolution of the rust layer. Characterization of the corrosion products was performed using a portable microscope, scanning electron microscopy (SEM), energy dispersive X-ray spectroscopy (EDS), and X-ray diffraction (XRD). The fitting analysis was conducted by combining the electrochemical measurement results and serves as a useful reference for the corrosion behavior analysis and life prediction of ER8 wheel steel in neutral chloride environments.

## 2. Materials and Methods

The material used in this work was ER8 wheel steel, and the chemical compositions are shown in Table 1. For this work, the specimens were cut into coupons with dimensions of 20 mm × 20 mm × 5 mm and 10 mm × 10 mm × 10 mm. Four parallel specimens of dimensions (20 mm × 20 mm × 5 mm) were used for each immersion period in the weight loss experiments and to characterize the morphology of the rust layer formed on the steel surface, whereas specimens of dimensions (10 mm × 10 mm × 5 mm) were used for electrochemical analysis. Before use, these specimens were first cleaned ultrasonically with ethanol, then rinsed with distilled water, dried, weighed and stored in moisture-free desiccators.

**Table 1.** Chemical compositions of ER8 wheel steel (mass fraction/%).

| C | Si | Mn | P | S | Cr | Fe |
|---|---|---|---|---|---|---|
| 0.55 | 0.40 | 0.80 | 0.020 | 0.015 | 0.30 | Bal. |

The total immersion test was divided into two parts, one with the test period as the independent variable and the other with the chloride salt solution concentration as the independent variable, to analyze the effect of concentration on the corrosion behavior of the specimens. For the gravimetric experiments, corrosion products formed on the retrieved specimens were removed chemically by immersion in a specific solution (500 mL HCl + 500 mL distilled water + 3.5 g hexamethylenetetramine) and stirred for 10 min. After the corrosion products were removed, the specimens were rinsed with distilled water, dried with cold air and weighed to determine their respective weight losses.

The corrosion rate (CR), from the mass loss, can be calculated (according to ASTM Standard [17]) using the following Equation:

$$CR = (K \times W)/A \times T \times D \tag{1}$$

where CR is corrosion rate (mm/year), $K$ is a constant equal to $8.76 \times 10^4$, $W$ is the mass loss (g), $A$ is the surface area ($cm^2$), $T$ is the soak time (h), and $D$ is the density of ER8 wheel steel ($g/cm^3$).

Electrochemical measurements were performed with a CHI-660e electrochemical workstation at room temperature. The electrochemical measurement was carried out using

a three-electrode cell with platinum as the counter electrode, a saturated calomel electrode (SCE) as the reference electrode, and a specimen with an exposed area of 1.0 cm$^2$ as the working electrode. The test solution used in all experiments was a NaCl solution prepared from reagent grade chemicals and distilled water.

For the polarization measurements, anodic and cathodic polarization measurements were separately performed on different specimens at a scan rate of 1 mV/s. The scan range for the cathodic polarization measurement was an open-circuit potential of −500 mV relative to the open-circuit potential, whereas the scan range for the anodic polarization measurement was an open-circuit potential of +500 mV. The electrochemical impedance spectroscopy (EIS) measurement scanning frequency was in the range of 100,000 to 0.1 Hz, with a sinusoidal alternating amplitude of 5 mV. The data were fitted with Zview2. Prior to all electrochemical measurements, the specimen was left to soak freely in solution until the open circuit potential stabilized. Each electrochemical test was repeated three times to guarantee data reproducibility.

The macroscopic and microscopic morphologies of the specimens after corrosion in the solution at different corrosion development periods were observed by optical microscopy (OM) and scanning electron microscopy (SEM). The composition and distribution of the main elements in the rust layer on the surface were detected by EDS. Rust was scraped from the rusted specimen and analyzed by X-ray diffraction (XRD) to determine the phases. The XRD measurements were carried out using a PANalytical X 'Pert powder XRD diffractometer with a Cu target operating at 40 kV and 40 mA, a 2θ = 10°–90° range and a scanning speed of 2° min$^{-1}$.

## 3. Results and Discussion

### 3.1. Determination of the Corrosion Rate

The average corrosion rate determined using Equation (1) under different test conditions is shown in Figure 1. The average corrosion rate of ER8 wheel specimens in 2.0% NaCl solution shows a linear increase with increasing corrosion time and reached a maximum of 0.143 mm/year at 2160 h. In a 3.5% NaCl solution, the average corrosion rate first decreased (in stage I and stage II), then increased gradually (in stage III), and it reached the maximum value (0.129 mm/year) at 2160 h. In 5.0% NaCl solution, the corrosion rate showed an obvious wavy pattern of change in stage I and stage II and increased with increasing soaking time in stage III, and the maximum corrosion rate was 0.131 mm/year. With the gradual increase in the experimental period, the final corrosion rate of the experiment shows a pattern: 2.0% > 5.0% > 3.5%. In general, the experiment time has a greater effect on the corrosion rate of specimens at a lower concentration. The solution was changed every 7 days, and the corrosion rate was lower at 2.0% and 5.0% than at a 3.5% concentration in the initial stage due to the combined effect of solution concentration and dissolved oxygen content [18]. On the one hand, increasing the concentration of the solution will lead to an increase in the conductivity of the solution, accelerating the corrosion process; on the other hand, the solution concentration continues to increase but will reduce the content of dissolved oxygen and lead to an increase in the concentration of $Fe^{2+}$ near the anode, thereby increasing the resistance to electron transfer, inhibiting the reaction, and resulting in a reduction in the corrosion rate. Later, surface corrosion products were gradually generated by accumulation, but due to the presence of $Cl^-$, pitting corrosion was easily triggered, reducing the area of the protective corrosion product layer; the reduction in the amount of corrosion products is gradually involved in the corrosion process, causing corrosion to occur further, thus increasing the corrosion rate [19,20].

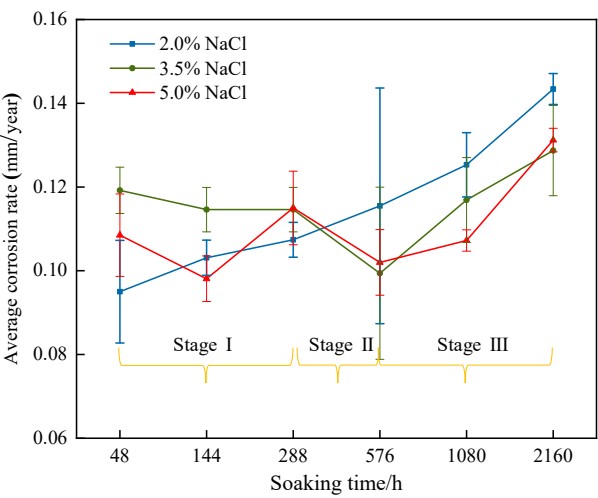

**Figure 1.** Average corrosion rate of ER8 wheel steel.

### 3.2. Analysis of the Corrosion Morphology and Composition

The macroscopic surface morphology of ER8 wheel steel before and after the test under different immersion conditions is shown in Figure 2. The rust layers on the surface of the ER8 wheel steel showed remarkable changes with time, from the early yellow-brown color, to reddish-brown, and finally to tawny. The coverage area of the surface rust layer gradually increased, the rust layer became denser, and the surface morphology significantly changed [21]. As shown in Figure 2a–c, there were significantly more corrosion products on the surface of the specimens soaked for 288 h than on those soaked for 48 h; additionally, the surface corrosion products began to peel off. The corrosion products basically covered the surface of the substrate after soaking for 144 h, as shown in Figure 2b,d,e. With an increasing NaCl concentration, the surface layer of the specimen gradually peeled off, which exposed some black areas on the surface of the specimen. With increasing immersion time or concentration of the NaCl solution, the coverage area of the corrosion products dominated by iron oxides gradually increased, and the corrosion products on the surface became denser.

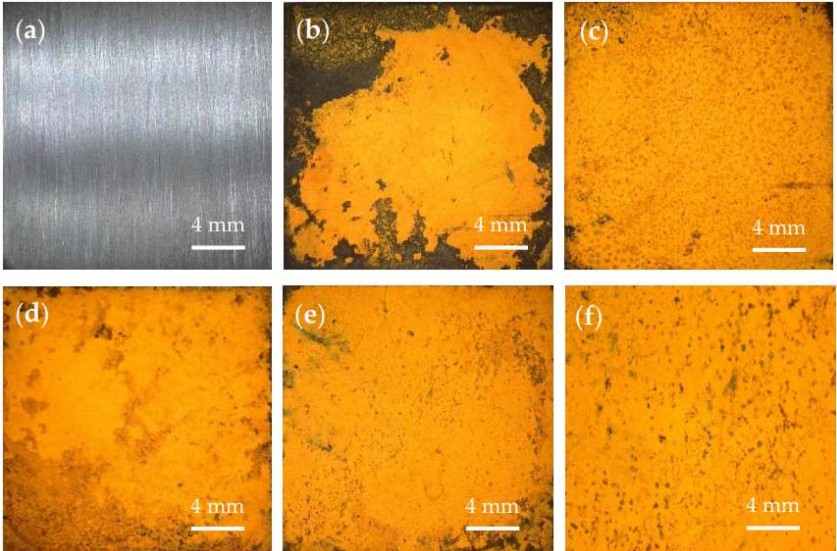

**Figure 2.** Macroscopic morphologies before and after corrosion test: (**a**) before corrosion test; (**b**) 48 h in 3.5% NaCl; (**c**) 144 h in 3.5% NaCl; (**d**) 288 h in 3.5% NaCl; (**e**) 144 h in 2.0% NaCl; and (**f**) 144 h in 5.0% NaCl.

The SEM morphologies of the corrosion products under different immersion conditions are shown in Figure 3. The results demonstrate that the surface corrosion products of the ER8 wheel steel specimen were mostly a cluster of flaky products. Figure 3a–c shows that there are many interstices between the initial corrosion products during the corrosion experiment, and these spaces were conducive to the penetration of corrosive media with the increasing corrosion time. The gaps between the clusters gradually decreased, and the rust layers at low and high magnifications were mostly lamellar structures. As the density of the surface corrosion products increased, cracks and holes appeared on the surface of the rust layers, which made the rust layer unable to protect the substrate from further corrosion. Figure 3b,d,e shows that the microscopic morphologies of the corrosion products of the ER8 wheel steel in different concentrations of NaCl solution are very similar, and the coverage rate and density of corrosion products in the 3.5% NaCl solution were the highest. The high-magnification image in the inset of Figure 3b illustrates the typical micromorphology of $\gamma$-FeOOH, which has a flowery microstructure [22,23].

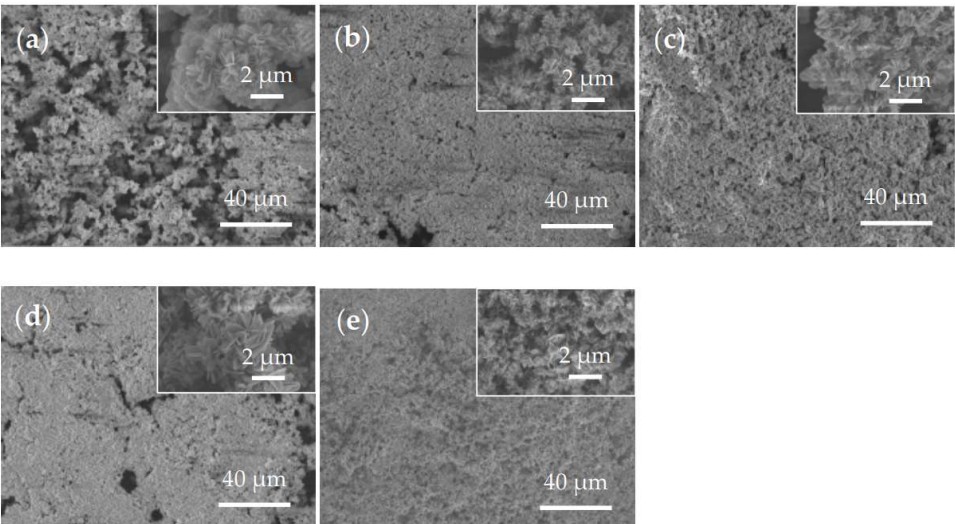

**Figure 3.** Surface morphology of the rust layer: (**a**) 48 h in 3.5% NaCl; (**b**) 144 h in 3.5% NaCl; (**c**) 288 h in 3.5% NaCl; (**d**) 144 h in 2.0% NaCl; and (**e**) 144 h in 5.0% NaCl.

According to Figure 4, the main components of the corrosion products under the different immersion conditions were $\alpha$-FeOOH, $\gamma$-FeOOH, and $Fe_3O_4$. The $\gamma$-FeOOH material has a feathery loose structure that facilitates the dissolution of the dissolved solvents. The results show that the experimental time had a significant effect on the composition of the corrosion product, whereas the concentration of NaCl solution had little effect on the composition of the corrosion products. Additionally, the corrosion products of the ER8 wheel steel immersed in different concentrations of NaCl solution were similar.

As is known, the permeability and mobility of oxygen in the NaCl solution is low, and the corrosion process is mainly controlled by the cathode process. Under the action of NaCl solution, the cathodic and anodic reactions occurring in the active zone in the initial immersion experiment corrosion test can be described by the following equations:

$$O_2 + 2H_2O + 4e^- \rightarrow 4OH^- \tag{2}$$

$$Fe \rightarrow Fe^{2+} + 2e^- \tag{3}$$

In an oxygen and water environment, $Fe(OH)_2$ was readily oxidized to $Fe(OH)_3$. The high corrosion rate resulted in oxygen depletion and induced the formation of $Fe_3O_4$. In the neutral solution, $\gamma$-FeOOH formed rapidly at the surface of the specimens. The continuous production of $\gamma$-FeOOH leads to an increase in hydrogen ion concentration and a decrease in pH. Under these conditions, $Fe^{2+}$ was suspended on the surface of $\gamma$-FeOOH, promoting

further dissolution of γ-FeOOH and conversion to α-FeOOH and Fe₃O₄. Combined with the SEM results of the corrosion product, γ-FeOOH first formed on the surface of the substrate and then converted into other oxides [18].

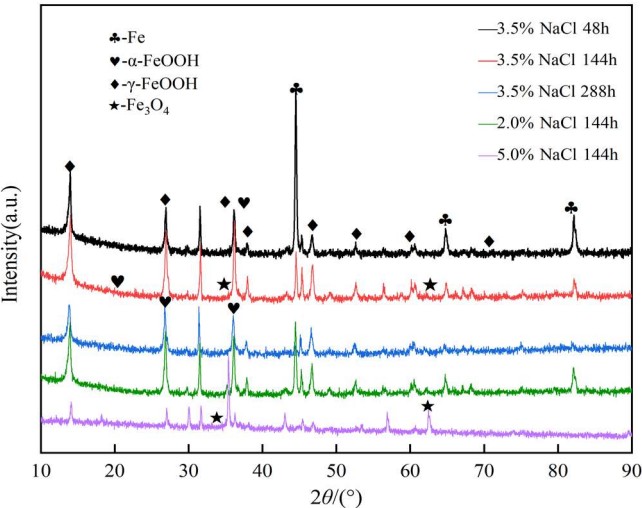

**Figure 4.** XRD patterns of the rust layers of ER8 wheel steel.

The initial corrosion products were less abundant and thin, and the amount of corrosion products gradually increased with corrosion time. At the same time, a large amount of γ-FeOOH was converted to α-FeOOH and Fe₃O₄, whereas γ-FeOOH was not stable and was eventually converted to α-FeOOH. α-FeOOH has the shape of a cotton ball and cotton wool, which have better adhesion. Since the crystals of α-FeOOH dendrites are slim, the crystal clusters that formed were dense, and the subsequent accumulation of dense crystal clusters makes the rust layer more resistant to corrosion ion invasion [24].

The SEM images of the specimen surface morphology after the removal of the rust layer are shown in Figure 5. The results show that the surface damage of the specimens after the removal of corrosion products was mainly characterized by spalling and pitting. In the initial stage of corrosion, the damage to the specimen consists mainly of a large number of dense corrosion pits. As the corrosion time increased, the area and depth of pitting gradually increased to the extent that adjacent pits were connected; thus, spalling occurred. Thus, after the corrosion products were removed, the surface of the sample showed large areas of pitting.

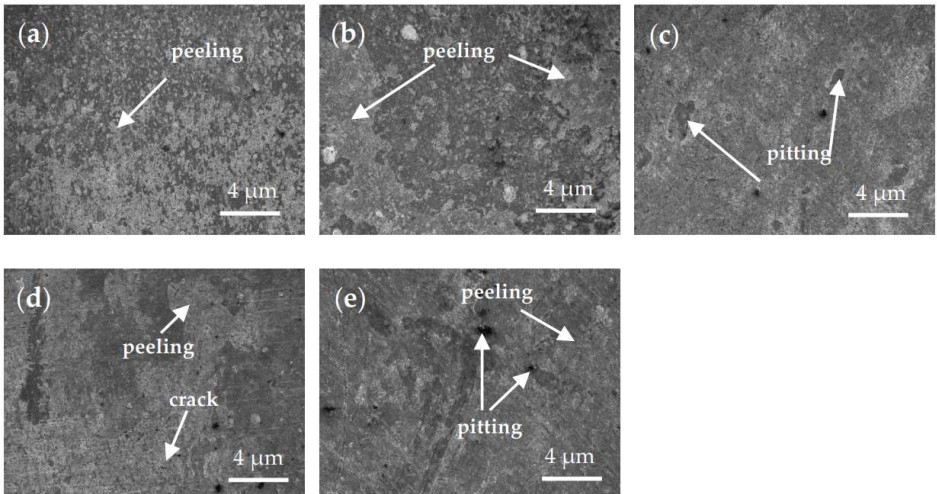

**Figure 5.** SEM images of the rust-removed surface: (**a**) 48 h in 3.5% NaCl; (**b**) 144 h in 3.5% NaCl; (**c**) 288 h in 3.5% NaCl; (**d**) 144 h in 2.0% NaCl; and (**e**) 144 h in 5.0% NaCl.

Figure 6 shows the cross-sectional morphologies of rust formed after the immersion experiment. With an increasing corrosion time, the thickness of the rust layer gradually increased, and cracks appeared in the cross-section of the rust layer (Figure 6a–c). With the continuous propagation of cracks, the corrosion products became loose and easily peeled off. Thus, the corrosive media could further contact and corrode the substrate material through these cracks, reducing the protective effect of the corrosion products on the substrate. Figure 6b,d,e shows that the thickness of the rust layer increases and then decreases with increasing solution concentration, but the rust layer is the densest in the 5.0% NaCl solution and has the best protective effect on the substrate. Figure 6c shows that corrosion products have a clear tendency to induce pitting, which coincides with the results of Figure 5c.

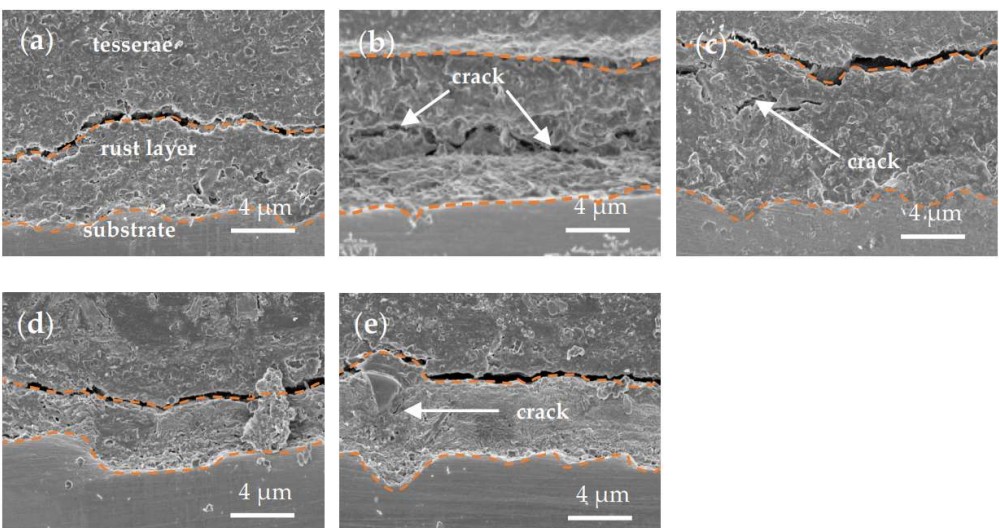

**Figure 6.** SEM images of the cross-section of the rust layer: (**a**) 48 h in 3.5% NaCl; (**b**) 144 h in 3.5% NaCl; (**c**) 288 h in 3.5% NaCl; (**d**) 144 h in 2.0% NaCl; and (**e**) 144 h in 5.0% NaCl.

Figure 7 shows the EDS results of the rust layer on the ER8 wheel steel surface. The results show that different test environments had little influence on the elemental content of the corrosion products. The EDS spectra of typical areas of the rust layer on the surface under different test conditions were obtained. The main elements in the corrosion products were Fe, Cl, C and O, indicating that the main components of the corrosion products under the different test conditions were iron oxides. In addition, a small amount of Si was found in the rust layers, either from the substrate itself or from impurities in the air. Cl atoms can penetrate the corrosion product layer, which is an important factor causing the pitting corrosion of the substrate. The content of elemental Cl in the rust layer obviously increases with increasing NaCl concentration. The elemental oxygen content represents the degree of corrosion of the substrate in the initial stage. The oxygen content increased and then decreased slightly with test time. Along with the increase in the NaCl solution concentration, the elemental oxygen content at 144 h was in the order 3.5% > 2.0% > 5.0%. This is similar to the results in Figure 1; increasing the concentration of the solution has an accelerating effect on the corrosion process. Although the effect of the increase in the solution concentration and immersion time on the corrosion product elemental content is relatively small, the combination of XRD and SEM results for corrosion products shows that the corrosion product composition changed.

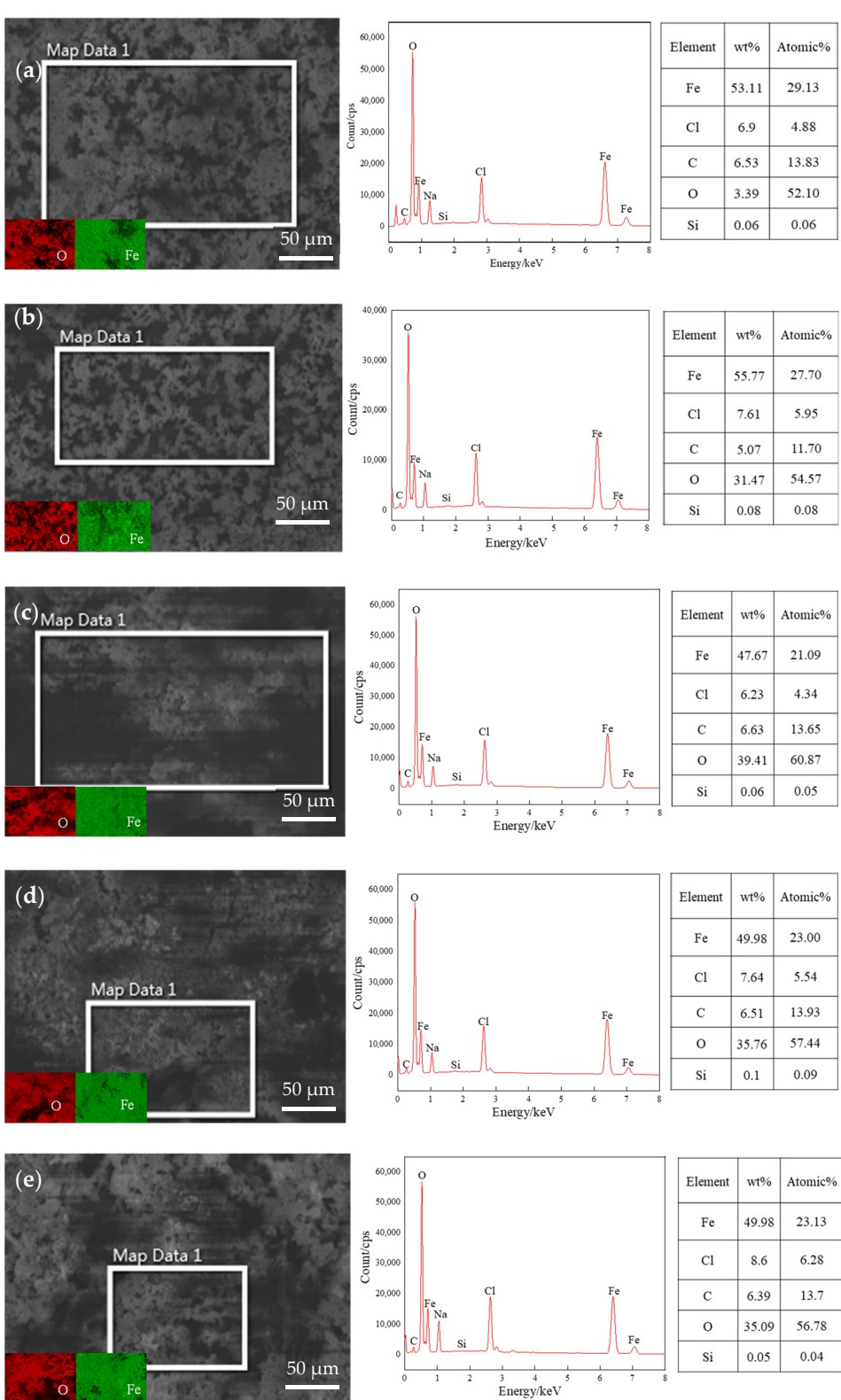

**Figure 7.** Energy dispersive spectroscopy images of the rust layer: (**a**) 48 h in 3.5% NaCl; (**b**) 144 h in 3.5% NaCl; (**c**) 288 h in 3.5% NaCl; (**d**) 144 h in 2.0% NaCl; and (**e**) 144 h in 5.0% NaCl.

*3.3. Electrochemical Analysis*

3.3.1. Open Circuit Potential Measurements

Figure 8 shows the OCP of ER8 wheel steel under different corrosion conditions. The results show that the OCP first increased and then decreased with the increasing NaCl

solution concentration and increased with the increasing immersion time. Furthermore, the OCP achieved a steady state when the immersion time was 600 s. The final OCPs of the ER8 wheel steel specimens were in the following order with respect to the immersion time: 288 h > 144 h > 48 h (Figure 8a). The final OCPs of the ER8 wheel steel specimens were in the following order with respect to the NaCl concentration: 3.5% > 2.0% > 5.0% (Figure 8b). It is generally believed that a lower OCP indicates higher electrochemical activity and more active corrosion with regard to carbon steel [19].

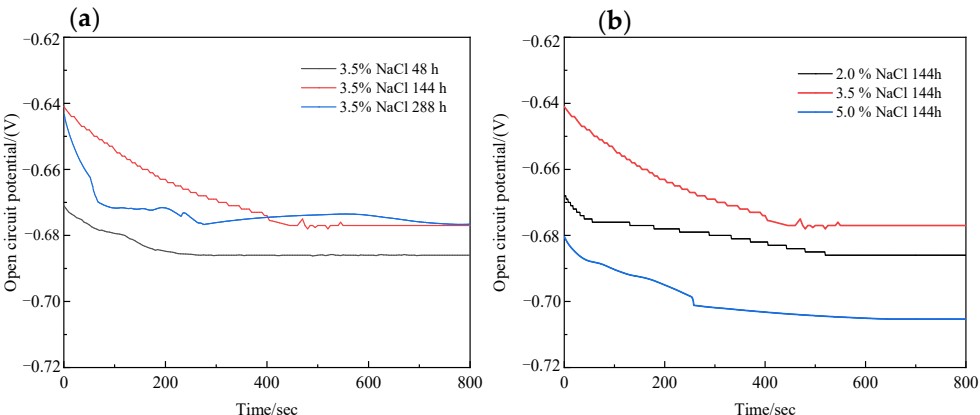

**Figure 8.** Open circuit potential of ER8 wheel steel under different conditions: (**a**) different immersion times and (**b**) different concentrations.

### 3.3.2. Measurement of Potentiodynamic Polarization Curves

The potentiodynamic polarization curves of ER8 wheel steel under different corrosion conditions are presented in Figure 9. The corrosion potential ($E_{corr}$) and corrosion current density ($I_{corr}$) of electrochemical specimens are calculated according to the Tafel extrapolation method, as summarized in Figure 9b,d. All steels demonstrated active corrosive behaviors, suggesting that a passive film did not form on the steel surfaces, and the main corrosion processes were dominated by cathodic reduction.

Figure 9a shows that with increasing test time, the $I_{corr}$ of the specimen first increased and then tended to be stable, and the $E_{corr}$ of the specimen first decreased and then tended to be stable. When the immersion time was 288 h, $I_{corr}$ reached a maximum of 46.72 $\mu A \cdot cm^{-2}$, and $E_{corr}$ reached a minimum of $-0.671$ V. The evolution of the cathodic process changed less with the increased soaking time, indicating the stabilization of the amount of reduced $\gamma$-FeOOH in the corrosion products. The corrosion current density ($I_{corr}$) of the specimen gradually increased, and the corrosion potential ($E_{corr}$) of the specimen continued to decrease as the concentration of the NaCl solution was increased (Figure 9c). The growth trend of $I_{corr}$ decreased with the increasing concentration of the NaCl solution. It shows that the change of $I_{corr}$ between 2.0% and 3.5% is more obvious than that between 3.5% and 5.0%. Among them, the $I_{corr}$ of the specimen after corrosion in 5.0% NaCl solution reached a maximum of 42.48 $\mu A \cdot cm^{-2}$, and the $E_{corr}$ of $-0.660$ V was the smallest. As expected, the anodic branches of the oxidation trends of the specimens positively shifted with an increased soaking time and NaCl concentration, which also meant that $I_{corr}$ increased when the soaking time and NaCl concentration were increased, showing an almost linear trend.

$I_{corr}$ belongs to the category of kinetics and is related to the corrosion rate, which should be considered first when evaluating corrosion resistance. The corrosion rate decreases and the corrosion resistance improves with decreasing $I_{corr}$. Thus, the specimens with rust layers have worse corrosion resistance when the immersion time is extended. $E_{corr}$ is a thermodynamic concept that reflects the potential for corrosion, and the dynamic characteristics of materials cannot be ignored [25]. The corrosion potential determines the specific formation of the passive film and its consequent stability in the NaCl solution. That is, increasing immersion time and NaCl solution concentration will reduce the corrosion resistance of the specimen.

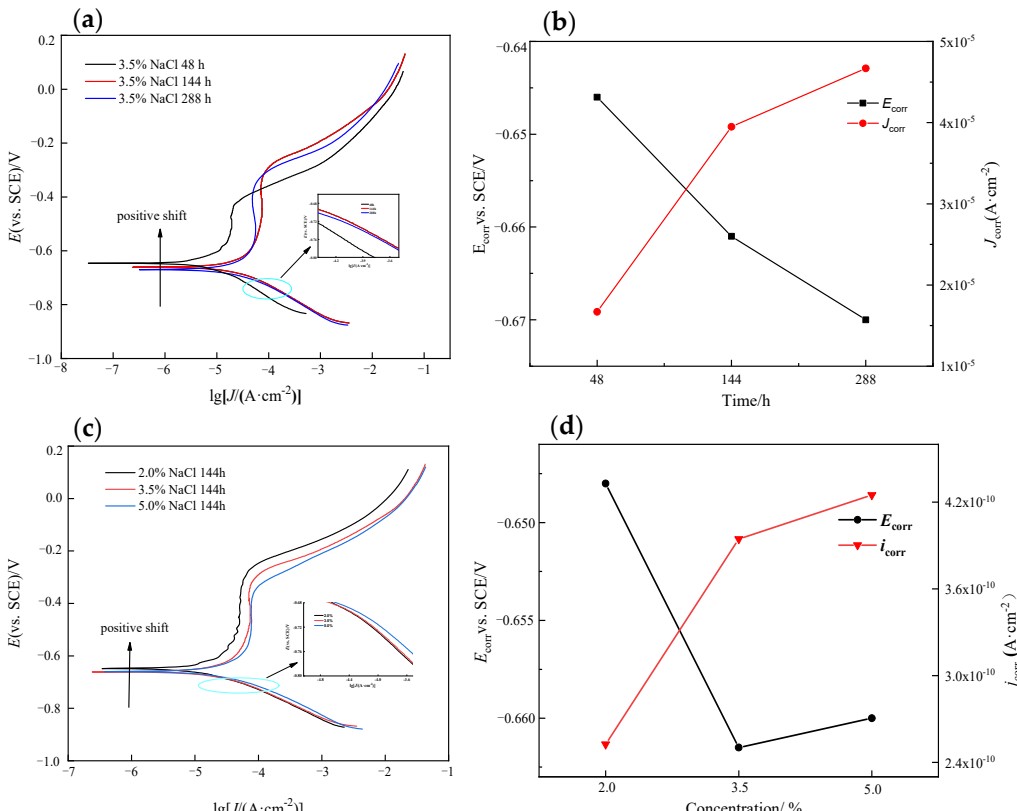

**Figure 9.** Potentiodynamic polarization curves of specimens under different conditions: (**a**) different immersion times; (**b**) fitting results with immersion time; (**c**) different concentrations; and (**d**) fitting results with concentration.

Furthermore, $Cl^-$ effectively promotes rust layer breakdown and accelerates the electro dissolution of carbon steel [26,27]; therefore, an increase in the concentration of NaCl solution promotes the further corrosion of the rusted specimen. $I_{corr}$ increased with increasing $Cl^-$ concentration, which was similar to the mechanism of catalytic electrolysis due to $Cl^-$ [27].

### 3.3.3. Electrochemical Impedance Spectroscopy Measurements

EIS measurement results were achieved to further analyze the corrosion behaviors of ER8 wheel steel. Figure 10 displays the Nyquist plots and corresponding Bode-impedance and Bode-phase plots of the electrochemical specimens. The corrosion resistance can be analyzed by looking at the radius size of the capacitance in the Nyquist diagram [28]. As seen in Figure 10a,c, the Nyquist plot shapes are all incomplete semicircle arcs, indicating that active corrosion processes occurred, and implying a similar corrosion mechanism under various test conditions.

As shown in Figure 10a, the Nyquist plots show that with an increasing immersion time, the radius of the capacitive reactance arc gradually decreased, indicating that the corrosion resistance of the electrochemical specimens gradually decreased. Generally, the impedance modulus of the low-frequency region in a Bode diagram reflects the charge transfer resistance, and the impedance modulus of the high-frequency region reflects the resistance of a rust layer. As shown in Figure 10c, with the increasing NaCl solution concentration, the impedance modulus in the low-frequency region decreases and then increases, and the impedance magnitude is the highest in 3.5% NaCl solution, which indicates that the resistance of the rust layer decreases and then increases with increasing solution concentrations.

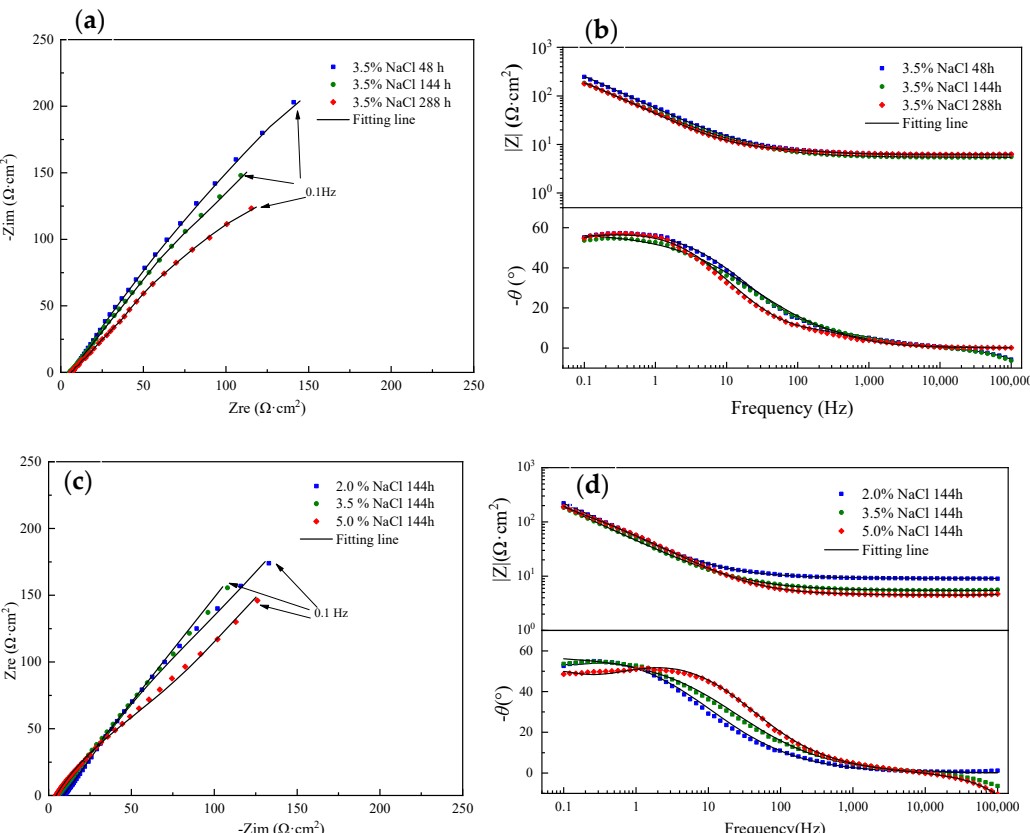

**Figure 10.** Nyquist plots and Bode plots of specimens tested under different conditions: (**a**) Nyquist plot with immersion time; (**b**) Bode plots with immersion time; (**c**) Nyquist plot with concentration; and (**d**) Bode plots with concentration.

The Nyquist plots in Figure 10a,c show a single capacitive resistance arc, so the equivalent circuit model shown in Figure 11 is established. The equivalent circuit is composed of solution resistance ($R_s$), a constant phase element (CPE), and charge transfer resistance ($R_t$). In this case, the capacitor was replaced with a CPE to improve the fitting quality, where the CPE included double-layer capacitance (C) and a phenomenological coefficient (n). A CPE is a useful modeling element with impedance given by the Equation (4) [28]:

$$\frac{1}{Z} = Q(jw)^n \tag{4}$$

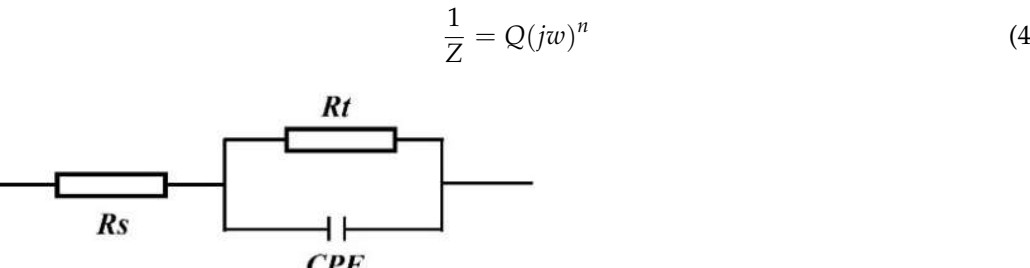

**Figure 11.** Equivalent circuit diagram based on the EIS results.

The CPE was used due to the uneven current density distribution or increased surface roughness (n). The fitting results obtained from EIS measurements of the electrical circuit models are summarized in Table 2. $Q$ and $R_t$ in the late stage of corrosion show decreasing and increasing trends, which indicates that the product layer gradually thickens and covers an increasing area. The corrosion resistance of the specimen gradually decreases with increasing immersion time.

**Table 2.** Electrochemical impedance spectroscopy fitting parameters.

| Condition | $R_s$ ($\Omega \cdot cm^2$) | $R_t$ ($\Omega \cdot cm^2$) | n | Q ($\Omega^{-1} \cdot cm^{-2} \cdot s^n$) |
|---|---|---|---|---|
| 48 h 3.5% NaCl | 6.18 | 252.68 | 0.82 | 0.00534 |
| 144 h 3.5% NaCl | 6.75 | 188.49 | 0.71 | 0.00579 |
| 288 h 3.5% NaCl | 6.37 | 163.96 | 0.63 | 0.00705 |
| 144 h 2.0% NaCl | 4.94 | 218.65 | 0.70 | 0.00598 |
| 144 h 5.0% NaCl | 9.11 | 194.36 | 0.68 | 0.00553 |

*3.4. Corrosion Mechanisms*

The present investigation is mainly concerned with the influence of immersion time and chloride salt concentration on the corrosion behavior and mechanism of ER8 wheel steel. SEM and XRD analyses, combined with the electrochemical test results, can be used to obtain the corrosion process diagram (shown in Figure 12).

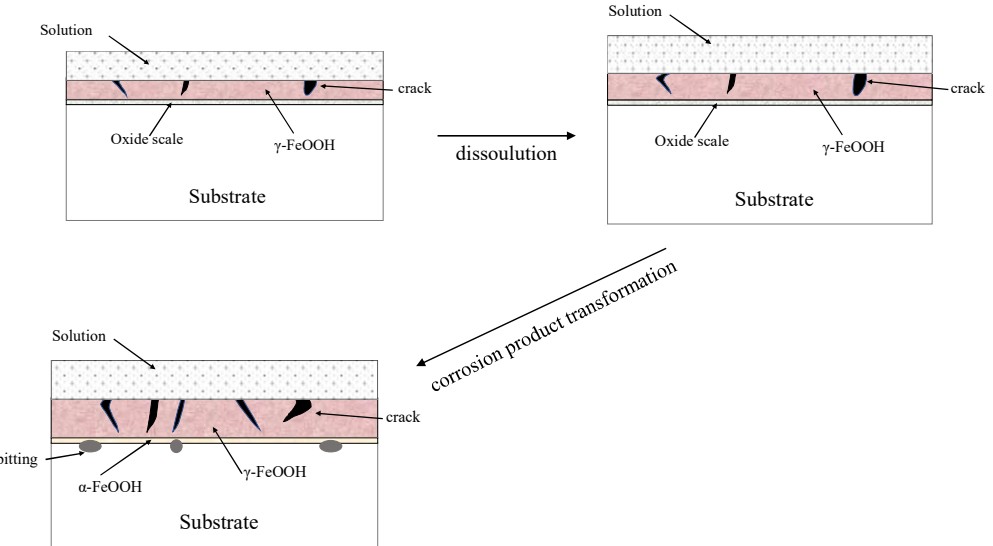

**Figure 12.** Corrosion mechanism of ER8 wheel steel in solution.

Note that the corrosion rate generally shows such a law: the initial test due to the substrate and solution being in full contact—and due to the high water-dissolved oxygen content—leads to a larger corrosion rate. This may be due to the substrate surface which is gradually covered by the corrosion layer, thus gradually decreasing the corrosion rate, and due to the continuously generated reduced γ-FeOOH participating in the corrosion process, leading to the further acceleration of the corrosion rate. The cross-sectional morphology shows that the corrosion layer cracks, and thus, more voids exist. Moreover, it is difficult to prevent $Cl^-$ from penetrating the corrosion layer, which means that the corrosion layer has difficulties when protecting the substrate.

The XRD results show that increasing the soaking time can alter the composition of the rust layer. Combined with the SEM results, the XRD results show that the rust layer is mainly composed of isomers of FeOOH, which are then converted into other oxides. The anodic reaction and cathodic reaction were mentioned above as Equations (2) and (3), and the total reaction is as follows:

$$Fe^{2+} + 2OH^- \rightarrow Fe(OH)_2 \tag{5}$$

According to Figure 3, $Fe(OH)_2$ is then oxidized to $Fe(OH)_3$, and $Fe(OH)_3$ is converted to γ-FeOOH and α-FeOOH. Although converted α-FeOOH is dense and can play a protective role, γ-FeOOH has a more destructive effect. Owing to the instability of γ-FeOOH, it

can be reduced with $Fe^{2+}$. The transformation of $\gamma$-FeOOH into $Fe_3O_4$ occurs according to the following reactions:

$$2\alpha\text{-FeOOH} + Fe^{2+} \rightarrow Fe_3O_4 + 2H^+ \tag{6}$$

$$2\gamma\text{-FeOOH} + Fe^{2+} \rightarrow Fe_3O_4 + 2H^+ \tag{7}$$

Although the thickness of the rust layer increases with immersion time, the degree of protection provided by the rust layer increases, and the reducing substances in the corrosion products accelerate further corrosion. This can also be verified by the results of polarization curve fitting (Figure 9b); with an increasing immersion time, $I_{corr}$ increases.

After rust removal, SEM showed pitting corrosion, a very localized form of corrosion, which is mainly related to $Cl^-$. This is an autocatalytic corrosion process; the metal in the small holes is dissolved, and the concentration of $H^+$ in these holes increases. Although the reduction reaction of oxygen does not occur, the cathodic reduction reaction of oxygen occurs on the surface adjacent to the small holes so that the small holes expand along the depth direction; therefore, pitting corrosion is hidden and very destructive in $Cl^-$ media, causing easy pitting corrosion.

### 4. Conclusions

(1) The average corrosion rate of the ER8 wheel specimen in 2.0% NaCl solution shows a gradual increase over the whole corrosion cycle. Although the corrosion rate shows fluctuations at 3.5% and 5.0% concentrations before 576 h, the corrosion rate also shows a steady increase after 576 h. The dissolved oxygen and $Cl^-$ concentration have a more pronounced effect on the early corrosion behavior, which is uniformly reflected in the change in corrosion rate. Later in the experiment, reducing substances ($\gamma$-FeOOH) gradually participated in the corrosion process, and $Cl^-$ penetrated the corrosion layer, leading to significant pitting of the substrate and accelerating the corrosion process.

(2) The initial (48 h) corrosion products of ER8 wheel steel in the NaCl solution were mainly clusters and lamellar $\gamma$-FeOOH and a small amount of $\alpha$-FeOOH, both of which were converted to $Fe_3O_4$. The corrosion product composition changed as the experiment progressed; however, it did not change with changing the concentrations of NaCl solution.

(3) With the increasing immersion time and concentration of NaCl, $I_{corr}$ increases and $E_{corr}$ decreases. The increase in immersion time causes the corrosion resistance of the specimens to decrease, and the decrease is greater than the protective abilities of $\alpha$-FeOOH. In contrast, another corrosion product, $\gamma$-FeOOH, has an obviously accelerated corrosion effect, and removing the corrosion product morphology also resulted in obvious pits. Increasing the corrosion time and concentration of NaCl reduces the impedance of the specimen, thus accelerating the corrosion process.

**Author Contributions:** Conceptualization, methodology, C.-G.H. and Z.-B.S.; software, validation, R.-Z.Z. and Y.-Z.G.; formal analysis, investigation, data curation, Z.-B.S.; resources, writing—original draft preparation, C.-G.H.; writing—review and editing, J.-H.L.; formal analysis, visualization, Z.-B.X.; supervision, project administration, R.-W.Y. All authors have read and agreed to the published version of the manuscript.

**Funding:** This research was funded by the Guangdong Basic and Applied Basic Research Foundation, grant number 2019A1515110807, the Youth Innovation Talents Project of Guangdong Provincial Department of Education, grant number 2018KQNCX271, the Research Initiation Program for High-level Talents of Wuyi University, grant number AG2018001, and the Science and Technology Plan Project of Jiangmen, grant number 2021030101720004620.

**Institutional Review Board Statement:** Not applicable.

**Informed Consent Statement:** Not applicable.

**Data Availability Statement:** Not applicable.

**Conflicts of Interest:** The authors declare no conflict of interest.

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
