# Peer review of "Study on the Corrosion Behavior and Mechanism of ER8 Wheel Steel in Neutral NaCl Solution"

_coatings, doi:10.3390/coatings12050713_

Round 1

Reviewer 1 Report

This paper aims to study on the corrosion behavior and mechanism of ER8 wheel steel in neutral NaCl solution
The following comments are made: 
1.    The corrosion rate calculation method should be described in the Experimental section.
2.    Subsection 3.1: The authors claim that “The corrosion rates of the specimens in 3.5% and 5.0% NaCl solutions showed extremely similar patterns of variation”. This is not correct and should be revised. Although corrosion rate showed a wavy pattern of change in both 3.5% and 5.0% NaCl concentrations, at some points (soaking times) they do not overlap. Also, in addition to the final corrosion rate, the change of corrosion rate in different soaking times in specimens with different NaCl concentrations should be discussed.
3.     It is recommended to calculate and tabulate the amount of the different corrosion products in each condition.
4.    What do the red lines in Figure 6 show? If the red lines show the thickness of the rust layer, the reviewer does not agree with this conclusion “the thickness of the rust layer formed in the 3.5% and 5.0% NaCl solutions for the ER8 wheel steel are similar”. The thickness of the rust layer in Fig. 6b is obviously higher than those we can see in Figs. 6 d and e.
5.    Considering the micrographs in Fig. 7, by increasing the NaCl concentration, the amount of corrosion products on the surface first increases and then decreases with the maximum occurring at 3.5% NaCl concentration. This needs to be explained more. 

In short, I believe that this piece of research does not meet the quality standards of “Journal of Coatings”, and therefore do not recommend its publication in this Journal. The results have not been discussed correctly leading to incorrect conclusions about the relationship in the observations.

Reviewer 2 Report

Reviewer Recommendation and Comments for manuscript coatings-1695879 with the title: “Study on the Corrosion Behavior and Mechanism of ER8 Wheel Steel in Neutral NaCl Solution”, authors: C.-G. He, Z.-B. Song, Y.-Z. Gan, R.-W. Ye, R.-Z. Zhu, J.-H. Liu, Z.-B. Xu.

The authors present the corrosion behavior of ER8 wheel steel by immersion test and electrochemical polarization. The structure and morphology of the surface, after corrosion, were analyzed by optical microscopy, scanning electron microscopy and X-ray diffraction.

The main comments that I find useful for improving the quality of the article are presented below:

*line 114. ”where K is the corrosion rate (mm-a)”. This unit of measurement is unusual. What is ”a”?

*Figure 1. ”corrosion rare” y axis name

*line 158. ”Figure 2.”?

*Figure 8a. The concentration is 3.5%? It should be mentioned in the legend.

*General remark. The results shown in Figure 7 indicate comparable results. No clear distinction can be made between corrosion media / products. More explanations are needed.

*General remark. The results shown in Figure 9a,b (144/288) indicate comparable results. No clear distinction can be made between corrosion media / products. More explanations are needed.

*General remark. The corrosion mechanism is presented in a simplistic way, it needs to be much improved or deleted.

*The Coatings journal require a specific format of references, authors must pay more attention in their writing.

*There are some grammar and typing mistakes.

*The authors must revise the entire manuscript.

Reviewer 3 Report

The paper “Study on the Corrosion Behavior and Mechanism of ER8 Wheel Steel in Neutral NaCl Solution” is an interesting study about the effect of corrosion on ER8 Wheel Steel.  The effects of corrosion of different concentrations of NaCl solution are analyzed trough optical microscopy (OM) and scanning electron  microscopy (SEM), energy-dispersive spectroscopy (EDS) and X-ray diffraction (XRD). The scientific quality of the papers is good, and the topic is interesting. The scientific method appears correct, and the research activities are well programmed.

I suggest to accept the manuscript after minor revisions:

  1. I suggest removing in line 42 the precise economic loss amount or to insert the reference year.
  2. Please insert a table of chemical composition of ER8 Wheel Steel
  3. Please insert in the legend figure 1 that the percentage are the concentration of NaCl Solution

Reviewer 4 Report

The authors have studied the corrosive behaviour of ER8 steel with different concentrations of NaCl solutions. The paper could be further improved by incorporating the following changes.

  1. Abstract has to be rewritten by incorporating obtained results and removing the repeated sentences (“Electrochemical results showed that the corrosion layer did not protect the substrate but accelerated the corrosion”).
  2. In section 3.1, provide the correct formula of corrosion rate by citing original reference. The formula is, Corrosion rate = (W1-W2) K / STD

         Here, K is constant (87600); mention the unit of corrosion rate is     mm/year instead of mm-a  (also in Fig 1 legends).  Also provide the clear image, the line colours used are not clearly visible in  Fig 10 too.

  1. It is better to provide Samples photographs (before and after corrosion test) prior to the morphology images.
  2. In Fig 3, SEM images micron markers are shifted. Use uniform pattern for all images.  Try to avoid the figure legends in white background [of (a), (b)…]   
  3. In Fig 4. XRD: Provide the JCPDS card numbers. Some peaks are unassigned (Eg. around 31 degree etc), assign the same.

   Give the reason for shifting the major peaks (around 38, 45 degrees etc.) at 144h 5.0% condition.

  1. Similarly, Fig 5 legends have careless mistakes, due to this the representation of pitting, peeling and crack regions are misplaced. Align properly with uniformity.
  2. In Fig. 7 of EDS spectra, the given elemental values are weight % or atomic %, provide both the values. Where is the elemental % of Cl? It should be included in the figure and accordingly at the text portion      with proper description.  Also assign all the peaks (C, Si etc).
  3. Page No. 8, Line 250: Correct the immersion time: 288h>144h>48h
  4. The description on Figure 9 should be elaborated in the manuscript.
  5. Conclusion part should be rewritten, the existing one is not complete and precise.
  6. English usage throughout manuscript must be substantially improved.

Round 2

Reviewer 1 Report

The manuscript can be published in the present format.

Author Response

We are very grateful for your comments on the manuscript. We have carefully considered the comments and revised the manuscript accordingly.

Reviewer 2 Report

Reviewer Recommendation and Comments for manuscript coatings-1695879 with the title: “Study on the Corrosion Behavior and Mechanism of ER8 Wheel Steel in Neutral NaCl Solution”, authors: C.-G. He, Z.-B. Song, Y.-Z. Gan, R.-W. Ye, R.-Z. Zhu, J.-H. Liu, Z.-B. Xu.

The authors responded to the reviewers' comments. Even under these conditions, the authors must correct certain mistakes such as:

*the corrosion rate, according to equation (1) must be expressed in cm/h. In the Results and Discussions section, the corrosion rate is expressed in mm/year. This means that either the equation is wrong or the experimental values are wrong.

*”Figure 8. Open circuit potential of ER8 wheel steel under different conditions: (a) different immersion times and (b) different concentrations.”?!? Figure 8a and 8b are identical!!!

*please check reference format for: 6-capital letters, 12, 14-journal abbreviation, 13-authors et al, 24-capital letters

Author Response

(The authors gave the same response as above.)

Reviewer 4 Report

The following queries were not addressed

1. It is better to provide Samples photographs (before and after corrosion test) prior to the morphology images.

Authors have provided the micrographs. Along with micrographs, asked the original sample photographs before and after corrosion test

2. In Fig. 7 of EDS spectra, the given elemental values are weight % or atomic %, provide both the values.

Asked to provide both the values weight % and atomic %

Author Response

(The authors gave the same response as above.)
